# Do Reactive Oxygen and Nitrogen Species Have a Similar Effect on Digestive Processes in Carnivorous *Nepenthes* Plants and Humans?

**DOI:** 10.3390/biology12101356

**Published:** 2023-10-23

**Authors:** Urszula Krasuska, Agnieszka Wal, Paweł Staszek, Katarzyna Ciacka, Agnieszka Gniazdowska

**Affiliations:** Department of Plant Physiology, Institute of Biology, Warsaw University of Life Sciences-SGGW, 02-776 Warsaw, Poland; urszula_krasuska@sggw.edu.pl (U.K.); pawel_staszek1@sggw.edu.pl (P.S.); katarzyna_ciacka@sggw.edu.pl (K.C.); agnieszka_gniazdowska@sggw.edu.pl (A.G.)

**Keywords:** carnivorous plants, *Nepenthes*, reactive nitrogen species, reactive oxygen species

## Abstract

**Simple Summary:**

Traps of *Nepenthes* (pitcher plants), often referred to as an “external stomach”, conduct a unique process of external digestion. In this process in pitcher plants, reactive oxygen species (ROS) primarily act positively, whereas they are considered as “evil characters” in humans. Reactive nitrogen species (RNS) have a dual role, which depends on their concentration and the place of their generation. The digestive process in *Nepenthes* is influenced by reactive oxygen and nitrogen species (RONS), which serve as regulators in both plant and human systems.

**Abstract:**

Carnivorous plants attract animals, trap and kill them, and absorb nutrients from the digested bodies. This unusual (for autotrophs) type of nutrient acquisition evolved through the conversion of photosynthetically active leaves into specialised organs commonly called traps. The genus *Nepenthes* (pitcher plants) consists of approximately 169 species belonging to the group of carnivorous plants. Pitcher plants are characterised by specialised passive traps filled with a digestive fluid. The digestion that occurs inside the traps of carnivorous plants depends on the activities of many enzymes. Reactive oxygen species (ROS) and reactive nitrogen species (RNS) also participate in the digestive process, but their action is poorly recognised. ROS and RNS, named together as RONS, exhibit concentration-dependent bimodal functions (toxic or signalling). They act as antimicrobial agents, participate in protein modification, and are components of signal transduction cascades. In the human stomach, ROS are considered as the cause of different diseases. RNS have multifaceted functions in the gastrointestinal tract, with both positive and negative impacts on digestion. This review describes the documented and potential impacts of RONS on the digestion in pitcher plant traps, which may be considered as an external stomach.

## 1. Carnivorous Plants

Carnivorous plants (*plantae carnivorae*) are among the most intriguing autotrophic organisms which influence human imagination. The question arises as to how this life form evolved and resembles a kind of mythological “mermaid” that entices animals, kills them, and digests their bodies. The increased interest in these plants is related to the substances synthesised in the digestive process. Some of these compounds are flavonoids or naphthoquinones, which could have potent medical uses as inflammatory therapeutics [1,2]. Pitcher-shaped trap leaves create specific environments that are rich in microorganisms producing enzymes with potential industrial uses, such as lipases resistant to low pH conditions [3].

The phenomenon of animal-eating plants began to be investigated by biologists almost two centuries ago. Darwin described them as “the most wonderful plants in the world” [4,5]. Carnivorous plants do not have one ancestor and do not belong to one systematic group. Carnivory, as an adaptation, evolved independently in the plant kingdom at least six times in different geographical locations [5]. These plants primarily evolved in moist environments with low nutrient availability. Their typical habitats are swamps or peat bogs with waterlogged soil that is poorly aerated, which are unfavourable conditions for most higher plants [6]. Carnivory involved reprogramming standard leaf physiology from assimilates donor organs to traps. These modified leaves enabled nutrient absorption from organic matter released in the trap. The transformation of assimilatory organs into traps was accompanied by the development of specific features that attract animals to the trap, including altering the leaf colour, generating UV patterns, emitting volatile compounds, producing nectar, and generating specific shapes. These unique heterotrophs are primarily photosynthesising autotrophs that digest animals and absorb nutrients from their bodies [7,8,9]. The plant carnivory syndrome hypothesis considers the ability of plants to attract prey into a trap, keep it there, kill and digest the prey, and absorb the released nutrients [9,10]. Some plants that had been considered as carnivorous do not achieve all of the properties of the carnivory syndrome. Some do not attract prey, and others are unable to digest trapped organisms by themselves [10]. These plants are often called protocarnivorous or semicarnivorous.

The carnivorous plant group is estimated to contain approximately 860 species [11]. Carnivorous plants are found on almost every continent except for very cold regions [12]. The majority of carnivorous plants have developed different methods of obtaining essential mineral nutrients (e.g., N, P, or sulfur (S)) [13]. Nutrient uptake in carnivorous plants depends on specific glands that secrete digestive enzymes and (or the same) glands that enable nutrient absorption [9]. Traps are commonly grouped into five categories: sticky, adhesive traps (fly-traps), pitcher-shaped containers, moveable snap traps, suction bladders, and eel traps [14].

## 2. Carnivorous Pitcher Plants

Carnivorous plants that produce a pitfall-formed trap are called “pitcher plants”, including Sarraceniaceae, Cephalotaceae, and Nepenthaceae [15]. The *Nepenthes* (Nepenthaceae) create jug-shaped traps that acquire arthropods and other small animals [16]. In this review the term “pitcher plants” is used exclusively for *Nepenthes*. Nepenthaceae are tropical plants that are primarily located in Indonesia, although some species occur in India (*N*. *khasiana* Hook. f.), Sri Lanka (*N. distillatoria* L.), Seychelles (*N. pervillei* Blume), and Madagascar (*N. madagascarensis* Poir. and *N*. *masoalensis* Schmid-Hollinger) [17,18] (Figure 1). Pitcher plants grow in habitats of very low nutrient availability, such as heath forests (*kerangas*), peat swamp forests, and montane forests [19].

*Nepenthes* generate dimorphic traps on their leaf tips during ontogeny. The first trap type (terrestrial pitcher) rests on the ground and is commonly ovoid-shaped. The second trap type (funnel-shaped) is located above ground level and is known as an aerial pitcher [19,20]. The terrestrial pitcher trap is characteristic for young plants, which form compact rosettes and straight tendrils tipped with ovary pitchers. Mature plants are characterised by climbing stems with long internodes and curled tendrils, which enable attachment to surrounding plants [19]. Recently, a species of *N. pudica* [21] was also discovered that produces lower pitchers located only underground. Traps are present on wholly or partially achlorophyllous shoots. The pitchers resemble terrestrial pitchers in shape and structure [21].

*Nepenthes* leaves have similar morphologies, consisting of an extremely enlarged photosynthetically active leaf base called a lamina, and a tendril that carries the trap (Figure 2). Gravity is an important factor in the trapping process as an attracted prey falls into the hollow trap and has no way to climb out due to the trap structure [22]. The *Nepenthes* pitcher trap is divided into three functional parts (basic zones) [23]: a peristome (ribbed upper rim of the pitcher), a waxy zone [24], and a digestive zone [25] (Figure 2A). The trap lid is considered as a separate zone of the pitcher (the fourth functional zone), which primarily protects the trap against rain droplets [22,24] (Figure 2B). The top part of the trap is (usually) a hood-shaped appendix with glands that function to attract a potential prey. These glands are called extrafloral nectaries (EFNs) because they secrete sweet nectar and volatile compounds. EFNs also are located on the tendril [19]. The largest and most tightly packed glands may be present on the peristome [19]. The peristome surface is generally wet and slippery, with inward-pointing hairs that cause the attracted prey to fall into the trap. The outside part is usually rough and hairy and possesses longitudinal rims to facilitate the animal access to the attractive peristome of the trap [22].

The upper part of the trap prevents the prey from climbing out because the inner surface is generally covered with downward-pointing hairs or loose wax crystals. The waxy zone contains modified stomata of hypertrophied guard cells (lunate cells) with curved surfaces that block climbing insects [24]. Insect escape is also made more complex by the anisotropic properties of the waxy zone of pitcher plants. The presence of moon cells causes insects to move at high speed and without innards towards the digestive zone, while their movement towards the peristome is difficult or even prevented [26]. The rough and non-cohesive surface of the wax crystals reduces insect adhesion to the waxy zone [25,27]. Even though the waxy zone plays such an essential role in reducing the adhesion of insects and preventing them from moving toward the trap opening, some pitcher plants, for example, *N. ampullaria* Jack and *N. bicalcarata* Hook.f. do not have this zone [28,29]. The released nutrients are adsorbed at the bottom of the trap, which is equipped with a permeable cuticle and glands (Figure 2C) that produce digestive enzymes. Transporters are located across the pitcher wall and actively transport liberated N into plant tissues [30]. The digestive fluid covers the lower part of this zone. The digestive fluid of pitcher plants, due to its unique physicochemical properties, plays not only an important role in digestion but also in the mechanical retention of the prey inside the trap [31]. In some species, e.g., *N. rafflesiana* Jack, the digestive fluid has viscoelastic properties that facilitate the prey drowning [19].

Several digestive enzymes and peptides have been identified in the pitcher fluid of various *Nepenthes* species [2,10,30,32,33]. The presence of thaumatin-like proteins belonging to pathogenesis-related proteins (PRPs) has been reported. PRPs inhibit the growth of microbial competitors (also fungi) in the digestive fluid [2]. Other compounds identified in the digestive fluid are: naphthoquinones, among others plumbagin and droserone, [33,34] and flavonols such as quercetin and kaempferol [35].

Some *Nepenthes* species differ in the morphology and functionality of the pitcher zones and in trapping mechanisms, such as *N. rafflesiana*, which exists in diverse ecological and morphological forms [19]. These alterations enable the examination of the capture strategy of carnivorous plants, by the development of plant–animal cooperation [19] as described below. Some pitcher plants have abandoned the typical “predator” system of nutrient acquisition. *Nepenthes* absorb N from different sources of animal or plant origin [36]. For example, *N. lowii* Hook.f. and other *Nepenthes* produce dimorphic pitchers. Typical terrestrial trap structures that capture arthropods are formed only by immature carnivorous plants. When the mature plant gains the ability to develop aerial pitchers, it changes the method of N uptake. Aerial traps are highly lignified with a modified lid and secrete buttery white exudates that attract the mountain treeshrew (*Tupaia montana*). This small mammal consumes nectar from the pitcher lid and excretes faeces into the pitcher. Nitrogen delivered by treeshew faeces accounts for 57–100% of nutrients absorbed by the plant, and is the primary source of this one macronutrient for leaves [36]. The elongated traps of *N. baramensis* utilise N captured from the faeces of Hardwicke’s woolly bats (*Kerivoula hardwickii*) [16,37]. These bats provide approximately 34% of N absorbed by the plant leaves [16,37].

## 3. *Nepenthes* Trap as a Human Digestive System

Despite the lack of similarity in the structure of the pitcher plant trap and the digestive system of animals, they perform comparable functions. A thorough description of the various digestive systems of carnivorous plants has recently been reviewed by Freund et al. [38]. The processes that occur in the pitcher plant’s regimen can be compared with the action of the human gastrointestinal (GI) tract. However, this is an oversimplification of the complex mammalian digestion. Nevertheless, the liberation of nutrients from an animal body in the pitcher bowl is similar to the softening of food in the GI tract (Figure 3). Food digestion and nutrient absorption in humans are mediated by defined components of the GI tract and digestive glands. The GI tract comprises the oral cavity, oesophagus, stomach, small intestine, large intestine, rectum, and anus. Nutrient release from food is mediated by the stomach and other GI compartments and juices secreted from salivary glands, pancreas, exocrine liver, and mucosal glands [39]. An undisturbed digestive process depends on other components of the GI tract, including smooth muscle activity, epithelial cells, and endocrine cells [39]. Carbohydrates, fats, and proteins are degraded by digestive enzymes such as amylases, lipases, endopeptidases, and proteases [40].

Human gastric fluid contains a high concentration of hydrochloric acid, which lowers its pH. High acidity maintains sterile conditions in the stomach and promotes the digestion of food proteins. The secretion of gastric acid is modulated by endocrine, paracrine, and nerve pathways, which mediate gastrin secretion, histamine and somatostatin release, and acetylcholine release, respectively. The inhibition of gastric acid secretion is under the control of the proton level in the gastric cavity [41]. Acid secretion is stimulated by histamine via the activation of the histamine H2-type receptors of parietal cells [42].

Stomach pH has been decreasing during three million years of human evolution. High stomach acidity was favoured as it disrupted pathogens, and stomach acidity is considered to function as an ecological filter [43]. The pH value of the stomach lumen is approximately 1.5 for a healthy adult (Figure 3), but higher for a premature infant (pH > 4) or an elderly human, which increases the risk of bacterial (pathogen) infection. Carnivores have a higher stomach acidity than herbivores [43]. This adaptive phenomenon is reflected in carnivorous plants. Young traps with closed lids and visually aged traps have a higher pH in the digestive fluid. Low pH in the digestive fluid is probably linked to the elimination of excess microorganisms and potentially harmful pathogens originated from the prey [44]. The digestion in the pitcher trap is probably highly dependent on the pH of the digestive fluid and the trap age. The digestive fluid of *N. rafflesiana* was monitored, and the pH value decreased rapidly during the first few hours after the trap opening (to a pH of around 4). This value continued to decrease over one week to around pH 2, although it was not induced by the prey capture [45]. Other studies reported that the application of nitrogenous solution (mimicking the prey input) to the digestive fluid or prey capture decreased the pH of the fluid [22]. The acidity of the digestive fluid is related to the species and the food from which the plant derives nutrients [46,47].

The digestive fluid pH may become more acidic as a result of the stimulation of the hydrolytic activity by the prey. The acidification of the digestive fluid in *N. ampullaria* is due to proton secretion by epidermal cells of the pitcher [48]. The plasmalemma H^+^-ATPase mediates the acidification of the pitcher fluid in *N. alata* Blanco [48]. The pitcher plant (*N. ampullaria*) also absorbs protons from the digestive fluid to prevent hyperacidity [48]. This regulation is an adaptive strategy that enables symbiotic bacteria and invertebrates to liberate N from the prey [30]. The authors demonstrated that after the pH is reduced to 4 in the pitcher trap, plants actively uptake protons in most zones of the pitcher to increase the pH of the digestive fluid. The lowest proton efflux rate was observed if the digestive fluid pH reached 6 [30].

Closed traps are sterile and contain only the plant-derived compounds [49], which facilitate the identification of specific proteolytic activities in pitcher traps. Protease activity during pitcher plant digestion can be detected and measured by conducting peptide-based fluorescence resonance energy transfer (FRET) [49]. Many hydrolytic enzymes which soften the prey and liberate nutrients have been identified in the trap fluid [50]. The protein identification of the secretome of closed and opened traps collected from *N. mirabilis*, *N. alata*, *N. sanguinea*, *N. bicalcarata*, and *N. albomarginata* confirmed the presence of many hydrolytic enzymes: aspartic proteases, glucanases, a cationic peroxidase, and class I, class III, and class IV chitinases, as well as a PR-1 protein and a thaumatin-like protein [51]. In addition to these proteins, the authors identified, and compared with the *Nepenthes* nucleotide database, peptides corresponding to new, putative digestive enzymes: another aspartic protease, serine carboxypeptidases of types 3, 20 and 47, α- and β-galactosidases, protein phosphatases, nucleotide pyrophosphatases/phosphodiesterases, new peroxidases of type 27 and type N, and lipid transfer proteins. Furthermore, the identification of the partial sequences pointed to additional putative enzymes: a serine carboxypeptidase, a β-xylosidase, an α-glucosidase, β-galactosidases, a β-D-1,3-glucosidase 7-like protein, and a lipase [51]. The best known nepenthesins I and II have been characterised in several *Nepenthes* species [52,53,54]. The acidic pH of the digestive fluid autoactivates nepenthesins [49]. These acid proteinases have a specificity to aspartic acid residues but the relationship with other aspartic proteinases is rather unclear [55,56]. In plants, under phosphate starvation or mechanical injury, self-incompatibility ribonucleases (S-like RNases) are induced. Moreover, in carnivorous plants, ribonucleases are highly expressed in the trapping organs [50,57]. The transcriptomic data of the pitcher different zones of *N. khasiana* indicated the presence of chitinases’ transcripts. The expression of *NkChit2b* was detected prior to the chitin induction, suggesting the participation of these transcripts as a constitutively expressed housekeeping protein [56]. In the digestive zone of the *N. khasiana* trap, high expression of a relatively large number of putative peroxidases was observed [56]. The authors claimed that these enzymes are the key components in the prey digestive machinery and protect against pathogen attack.

## 4. The Role of Reactive Oxygen Species in External Digestion by Carnivorous Plants

Reactive oxygen species (ROS) are products of the incomplete reduction or excitation of oxygen [58]. High ROS concentrations are primarily linked to inefficient antioxidant systems and lead to the induction of oxidative stress, which may result in cell death. ROS are necessary to perceive “normal” or “typical” functions of an organism because they act as crucial components of signalling transduction cascades [59]. At the physiological level, ROS participate in the removal of a toxic microbiome in animals and plants [60].

ROS include the superoxide anion (O_2_^•−^), hydroxyl radical (^•^OH), hydrogen peroxide (H_2_O_2_), and peroxyl, alkoxyl, and hydroperoxyl radicals [58]. In plant tissues, ROS are primarily produced in chloroplasts, mitochondria, peroxisomes, and the apoplastic space [61]. ROS are mainly formed in the Fenton and Haber-Weiss reactions [58,62]. The reactivity of ^•^OH is limited to nearby molecules, whereas H_2_O_2_ has a longer half-life and can be translocated to another cellular compartment [63,64]. The increased activity of the plasma membrane respiratory burst oxidase homolog (Rboh), a nicotinamide adenine dinucleotide phosphate (NADPH) oxidase, is linked to higher ROS generation [64]. ROS react with nucleic acids, proteins, lipids, and sugars [59], and are key factors of carcinogenesis in animals [65]. ROS attack on DNA structure results in single- or double-strand DNA breaks and DNA–protein cross-links [66]. Guanine oxidation at the C8 position leads to the formation of 8-hydroxy-2′-deoxyguanosine (8-OHdG) [67], which is mutagenic and related to mutations commonly observed in human cancers [65].

One of the most important modes of ROS action is their reactivity with amino acid residues in peptides/proteins, which leads to protein posttranslational modifications (PTMs). Carbonylation is an irreversible PTM occurring under physiological conditions [68]. Proteins with carbonyl groups formed on arginine, proline, threonine, or lysine residues usually lack normal function and are more prone to degradation [68].

The balance between ROS generation and decomposition is achieved by the antioxidant system, consisting of enzymatic and non-enzymatic components. The best known enzymatic ROS modulators are isoforms of superoxide dismutases, catalases, ascorbate peroxidases, glutathione peroxidases, and glutathione reductases. Thioredoxins, glutaredoxins, and peroxiredoxins are other proteins acting as ROS modulators [69]. Among non-enzymatic antioxidants are reduced ascorbic acid (ASA), reduced glutathione, carotenoids, and α-tocopherols [58,69,70].

The adverse effects of ROS in human digestion have been examined. ROS participate in tissue lesions during inflammatory processes [71]. Important sources of ROS include gastric epithelial cells and activated inflammatory cells such as neutrophils in infected tissues [72]. Carcinogenesis in the digestive system is strongly affected by ROS. Oxidative stress and ROS overproduction is coupled to *Helicobacter pylori* infection, one of the factors in cancer development [65]. The link between ROS generation in mitochondria and several environmental risks for the incidence of development gastric cancer has been proposed. ROS are implicated in gastric cancer invasion and metastasis [65]; their levels increase in chronic gastritis, inflammatory bowel diseases, and chronic liver diseases [73]. The results of animal studies indicate that gastroesophageal reflux stimulates ROS formation and leads to lesions of the oesophageal mucosa [74,75].

The consumption/digestion of meat (consisting of fat, proteins, and free and bound iron) by a human is strongly coupled to oxidative processes. Unstable ROS products of the Fenton reaction generate cytotoxic and genotoxic lipid oxidation products like malondialdehyde or 4-hydroxynonenal [76]. These compounds participate in protein carbonylation. Ageing tissues accumulate carbonylated protein aggregates [68]. ROS also contribute directly to protein degradation, especially of oxidatively modified proteins [77]. Protease activity in vitro is higher when the protein reacts with free radicals [78]. In *N. alata* Blanco pitcher fluid, the enzymatic digestion of products of the exogenous oxidised B chain of bovine insulin was observed [52]. Free radicals are assumed to accelerate protein breakdown, probably by their special modification (e.g., carbonylation) followed by degradation [68,77]. In plants, ROS-dependent changes in the activities of acidic proteases, chymotrypsin-like, peptidyl-glutamyl-peptidehydrolase, caseinolytic-specific, and trypsin-like proteases were reported in sugar-deprived maize (*Zea mays* L.) root tips [79], and germinating apple (*Malus domestica* Borkh.) embryos [80].

ROS are involved in the prey trapping and digestion in the carnivorous plant *N. gracilis* Korth. [81] (Figure 4). Electron paramagnetic resonance (EPR) spin trapping analysis demonstrated that the first step of the prey digestion is accompanied by the generation of free radicals (primarily ^•^OH) in the pitcher fluid. To prove that ROS induce protein degradation in the trap digestive fluid, myosin (an abundant protein in insect bodies) was added to the fluid of a young, closed trap along with a cocktail of protease inhibitors, and gel electrophoresis of the fluid protein fraction was performed. Myosin light chains were degraded, indicating that free radicals are beneficial for prey digestion [81]. In the *N. ventrata* Hort. ex Fleming digestive fluid, the presence of O_2_^•−^ was confirmed during the whole ontogeny of the trap, starting from the closed organs [44]. The authors proposed that radical forms of ROS are involved not only in the digestion per se but also play a role as compounds regulating the trap microbiome, although this needs further investigation.

## 5. The Significance of Reactive Nitrogen Species in Carnivorous Plant External Digestion

Reactive nitrogen species (RNS) include nitric oxide (NO) and compounds that are the products of its reaction with oxygen or O_2_^•−^. RNS are generated in different cellular compartments of plants and animals [82,83,84]. The nitrosonium cation (NO^+^), nitric oxide radical (^•^NO), nitroxyl anion (NO^–^), and peroxynitrite (ONOO^−^) and its protonated form—peroxinitrous acid (ONOOH)—are known as RNS [82,85]. RNS, particularly NO acting directly or indirectly, have regulatory/signalling functions in various metabolic processes [59,83,85].

In plants, NO synthesis is linked to enzymatic and non-enzymatic pathways or oxidative and reductive pathways. NO is generated by nitrate reductase, mostly the NIA1 isoform, as demonstrated in *Arabidopsis thaliana* ((L.) Heynh.) [86]. Molybdenum cofactor (Moco)-containing enzymes are proposed to be involved in NO generation [87]. In mammalian cells, the main NO-producing enzyme is NO synthase (NOS). This bimodal enzyme is present in three isoforms: constitutive neuronal NOS (nNOS), endothelial NOS (eNOS), and inducible NOS (iNOS) that catalyse the release of NO and citrulline from arginine in the presence of oxygen. This reaction requires many cofactors, such as NADPH or flavin mononucleotide [88]. In higher plants, the existence of a NOS-like enzyme is still not proven [89], but its activity (a reaction requiring all cofactors and co-substrates of mammalian NOS) has been measured [84]. The failure to identify arginine-dependent and oxygen-dependent NO-producing enzymes in higher plants has caused some authors to use the term “nitric oxide generating (NOG)” instead of NOS-like enzyme [90]. One well-described autotrophic organism with a confirmed NOS protein is a green alga, *Ostreococcus tauri* [88].

Nitrite (NO_2_^−^) is the main non-enzymatic RNS donor. Low pH stimulates the protonation of NO_2_^–^ to nitrous acid (HNO_2_). The further formation of different nitrogen oxides (NO_x_) depends on the redox state of the local environment [91,92]. ASA and other reductants increase the rate of NO release from NO_2_^–^ [91].

RNS like ROS modify cellular compounds such as nucleic acids, fatty acids, and proteins. The best-known RNS-initiated PTMs of proteins are *S*-nitrosation and nitration [93]. Similarly to the formation of carbonyl groups in proteins, tyrosine residue nitration is commonly accepted as an irreversible modification [94,95,96]. Nitro-tyrosine-containing proteins are also believed to be degraded more rapidly as observed for carbonylated proteins.

RNS are generated in various compartments of the human GI tract, including the salivary glands, oesophagus, stomach, small and large intestine, pancreatic glands, and liver. NO has several physiological and pathophysiological roles [39]. The stomach lumen is considered as the main source of NO due to the low pH and high NO_2_^–^ concentration [92,97]. In the oral cavity, nitrate (NO_3_^−^) is reduced to NO_2_^−^, which in the gastric compartment is converted into NO. This process is accelerated in the presence of reductants, e.g., dietary polyphenols [98,99]. NO in the stomach promotes PTMs, particularly the nitrosation of gastric mucosal proteins [99]. Stomach pepsinogen and occludin undergo nitration, which modifies their biological function [97,98]. For example, the nitration of pepsinogen resulted in less efficient protease (pepsin) formation, associated with the prevention of acute peptic ulcer disease [98]. It is commonly believed that NO functions as a defence molecule against microbial pathogens. In mammals, NO also may serve as a neurotransmitter in the regulation of blood flow and saliva secretion, and may control GI motility. NO impacts blood flow in most organs of the digestive system, including the stomach, appears to regulate water secretion and exocrine pancreatic secretion [39], and is proposed to protect the stomach against injury caused by stress conditions, HCl, or ethanol [39]. NO may act as a toxic compound in cancer development due to reactions with amines and imides and the formation of N-nitroso compounds (e.g., nitrosamines), which have carcinogenic effects [100,101]. Nitrite salt is used in meat processing to inhibit *Clostridium botulinum* and induces the pink colour of meat [76]. In the acidic stomach environment, HNO_2_ forms dinitrogen trioxide (N_2_O_3_), which is in equilibrium with NO and nitrogen dioxide (^•^NO_2_) [101]. Such specific conditions of the stomach favour not only protein nitration but also the nitration of fatty acids. Nitro fatty acids (NO_2_-FA) act as signalling molecules in animals and plants [102]. They can alkylate susceptible thiols of regulatory proteins and thus affect gene expression or the metabolic and inflammatory responses in animals. Fazzari et al. [103] demonstrated that olives and olive oil are sources of NO_2_-FA. Nitro-linolenic acid at a picomolar concentration was detected in Arabidopsis tissue (seeds, seedlings, and leaves) [102]. Generation of NO_2_-FA from the unsaturated fatty acids in the food under acidic gastric digestive conditions opens a new role for these redox-derived metabolites, probably typical for supporters of the Mediterranean diet.

RNS are present in *N. ventrata* Hort. ex Fleming digestive fluid [44]. The authors demonstrated that the highest NO production in the trap fluid (Figure 4) (using 4-amino-5-methylamino-20,70-difluorofluorescein diacetate, DAF-FM DA) was linked to digestion. Moreover, the main source of RNS is probably NO_2_^−^ whose concentration decreased after food application to the trap [44]. The presence of NO_2_^−^ and NO_3_^−^ in the digestive fluid, even in trace amounts, was demonstrated for *N. alata* pitchers [104] and *N. ventrata* [44].

RNS physiological activity may be associated with modulation of the ROS level (Figure 4). To underline the interaction of ROS and RNS, a new nomenclature of RONS for both is proposed [105]. RONS modify protein structure that impacts proteolysis, as was proposed for *N. ventrata* [44]. RNS or NO-modified molecules may act as signalling agents during prey digestion in the traps of *Nepenthes* plants. At present, the totally unknown presence or role of NO_2_-FA in the digestive fluid or tissues of the trap may be a new scientific challenge, the explanation of which could change our knowledge on digestion in carnivorous plants.

## 6. Conclusions

*Nepenthes* pitchers are sometimes called an “external stomach”, as these plants perform external digestion of the prey in the trap lumen. The *Nepenthes* trap functionally corresponds to the human oesophagus, stomach, and intestines (Figure 3). The residual debris of digested animals is removed as the ageing pitchers are lost. Enzymes and other chemicals are secreted into the trap lumen to digest the prey. The digestion is affected by RONS, which function as regulators in both humans and plants. In pitcher plants, ROS primarily act positively, whereas they act negatively in the human GI tract or result from pathophysiological events. RNS have a dual role in the human digestive system, which depends on their concentration and the location of generation. RNS action is probably similar to ROS action in the digestion carried out in pitcher plants (Figure 4). However, due to limited experimental data, it is difficult to draw far-reaching conclusions and further investigations should be conducted.

## Figures and Tables

**Figure 1 biology-12-01356-f001:**
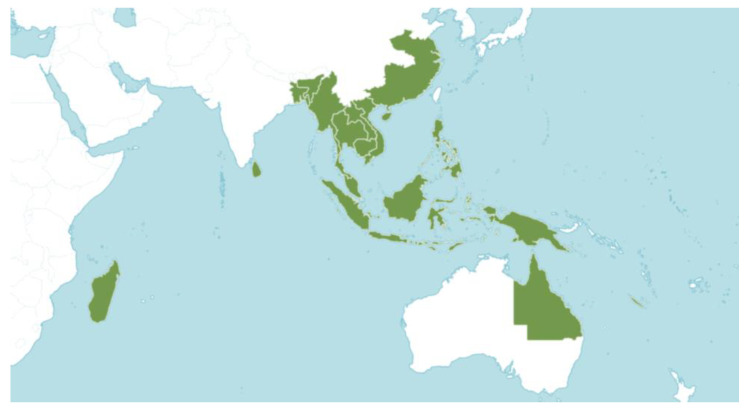
The worldwide range of the genus *Nepenthes*. POWO (2021) © RBG Kew.

**Figure 2 biology-12-01356-f002:**
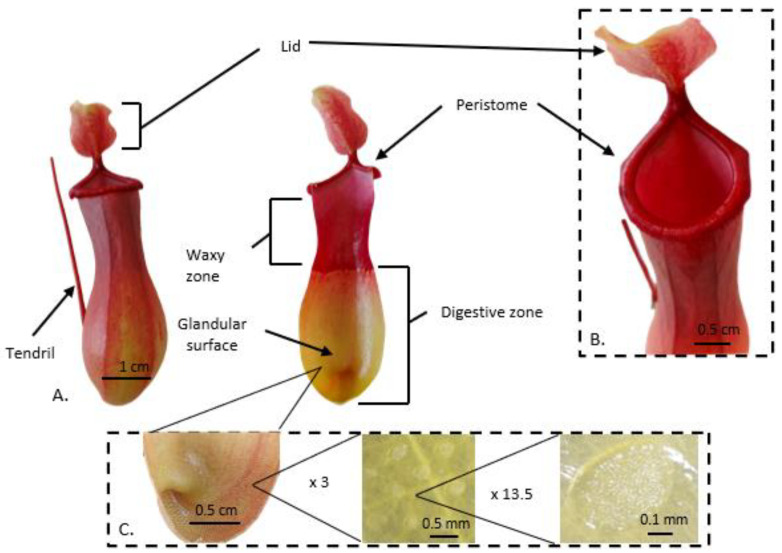
The aerial trap of the pitcher plant. (**A**) External and interior images show characteristics of the mature trap of *Nepenthes ventrata* Hort. ex Fleming (=*N. ventricosa* Blanco x *N. alata* Blanco). A lid is marked in the image of the whole (intact) trap tendril. (**A**) The peristome, glandular surface, and digestive zone are marked in the image of the trap cross-section. (**B**) Close-up image of the upper part of the trap. A peristome and a lid are marked. (**C**) shows images of the digestive glands at magnitudes of approximately 3× and 13.5×.

**Figure 3 biology-12-01356-f003:**
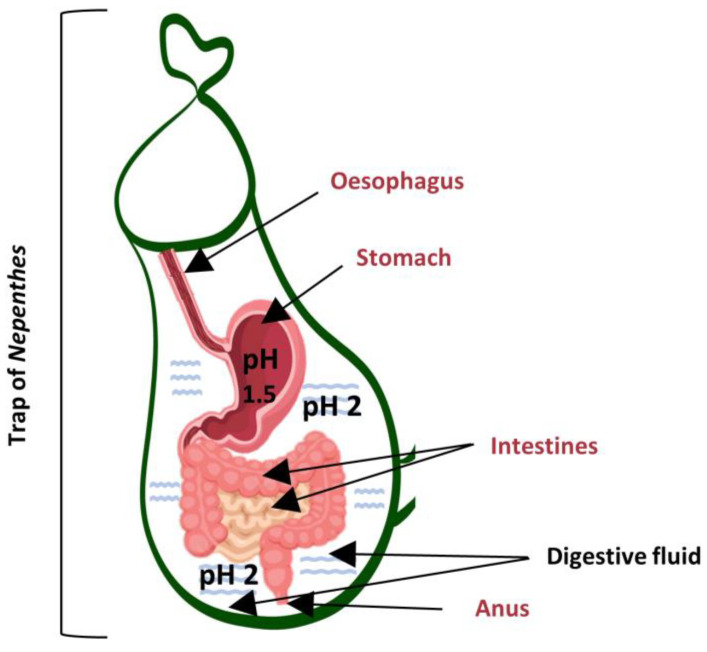
A diagram showing a *Nepenthes* trap as an organ corresponding to the human GI tract consisting of oesophagus, stomach, intestines, and anus. The value of the pH in the stomach is 1.5. A similar pH value, around 2.0, is in the digestive fluid of pitcher plants where the digestion of the prey is carried out. The trap has no human rectal-like area. In pitcher plants, undigested food debris is removed along with the ageing trap.

**Figure 4 biology-12-01356-f004:**
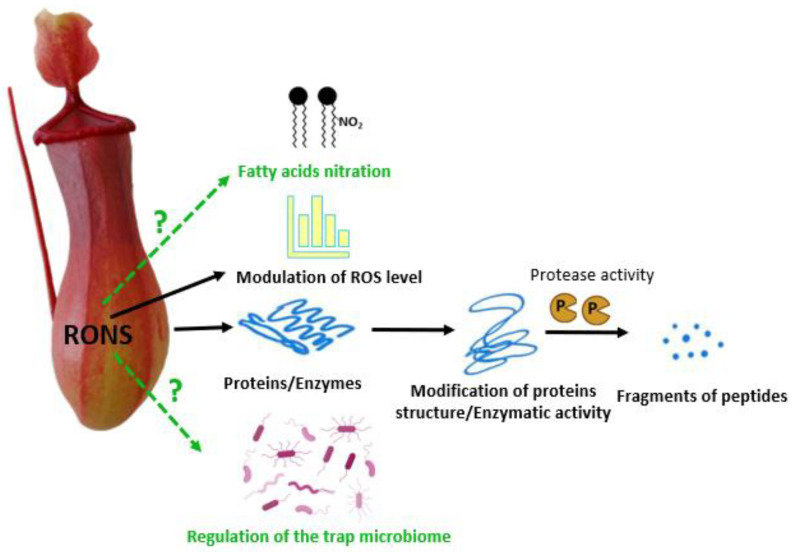
A scheme shows the role that RONS presumably perform (solid black lines) or may perform (dashed green lines with a question mark) in the digestion of prey by pitcher plants. RONS influence the activity of enzymes responsible for the prey decomposition, modify proteins, and facilitate their proteolysis. RNS modify the concentration of ROS in the digestive fluid. RONS alter cellular compounds, which may participate in fatty acid nitration. RNS may also have antimicrobial effects and control the growth of the microbiome in the digestive fluid in the pitcher plant’s trap.

## Data Availability

Not applicable.

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
