# Peer review of "Do Reactive Oxygen and Nitrogen Species Have a Similar Effect on Digestive Processes in Carnivorous Nepenthes Plants and Humans?"

_biology, 2023, doi:10.3390/biology12101356_

Round 1
Reviewer 1 Report
This review describes the documented and potential impacts of RONS (reactive oxygen species + reactive nitrogen species) on the digestion in pitcher plant trap, which may be considered as an external stomach. The article is very well structured, clearly written and supported by a very robust set of bibliographical references. The article gives us an excellent perspective on the anatomical, morphological, physiological, biochemical and molecular mechanisms of carnivorous plants, especially plants belonging to the genus Nepenthes (pitcher plants). The biochemical component is reviewed in great depth and the knowledge reviewed here has the potential to be very important for health and pharmaceutical sciences. The analogy of carnivorous plants “Nepenthes” with the human digestive system is also very well achieved. So, as a plant physiologist, I am in favor of publishing the article as it is.
Author Response
Thank you very much for such a positive review.
Reviewer 2 Report
The manuscript (biology-2644458) entitled “Do RONS modulate the digestion in pitcher plants?” by Krasuska et al. is submitted as a review to the Special Issue “Research of Nitric Oxide Signaling Molecules in Plants“. It focusses on the description of the digestive processes in mammals in comparison with that in the digestive fluid of carnivorous pitcher plants of the genus Nepenthes.
I appreciate that the authors put a lot of effort in this review but, unfortunately, I do not see any novelty in this paper. Recently, Freund et al. (2022, Plant Physiology 190:44-59) published a comprehensive review on the same topic including a comparison of digestion processes in mammals and carnivorous plants; Hedrich addressed the same some years earlier (2015, Current Biology 25, R100). In both cases, the authors addressed the carnivorous plant in general, not reducing themselves on one genus. The detailed comparison of mammalian digestion with that in Nepenthes is interesting but also sounds a bit far-fetched and much of it is textbook knowledge.
To be honest, the only two paper in the long reference list that are really related to the topic of the SI and the title of the paper are from Chia et al. (2004, Redox Report, 9, 255–261) and from the authors’ group (Wal et al., 2022, Plants 11, 3304). Both must be seen as preliminary studies; in particular, the Chia papers has never been confirmed within the last 20 years or any follow up paper is published, as far as I know. In addition, Wal et al. is in interesting study but the authors did not show any robust experimental data that justify any reliable connection to the digestive process. That means this review paper remains pure speculation based on only two thematically relevant but preliminary studies out of more than 100 references cited.
no comments, good to read
Author Response
We are sorry that the reviewer does not see anything new in our review work. However, we disagree with this comment. Work written by Freund et al. (2022) concerns the digestion by carnivorous plants with the impact on different methods of obtaining the prey. The manuscript discusses extensively the topic of digestive glands, types of traps, and the process of food hydrolysis, mainly proteolytic enzymes. A comparison of digestion in plants and humans is included but in a slightly different aspect than presented in our work. In the case of the work written by Hedrich (2015), it is difficult to talk about it as a review work (this is a “quick guide” entitled carnivorous plants). This work applies to various carnivorous plants. An image comparing the human digestive system is juxtaposed with a Venus flytrap. Our work concerns the digestion in the only one genus Nepenthes. The novelty of this work is the approach to the problem of ROS and RNS (RONS) in the aspect of food digestion and the comparison of how these molecules work in plants and humans. One of the premises of review works is also to include basic information (like in the textbook). In our opinion it is an advantage of the review not a disadvantage.
Two works strictly concern the subject of RONS in carnivorous plants. Many times, when writing works on one plant species, literature data for another are provided. This is not uncommon. We applied the same in our review work. We did not limit ourselves only to the involvement of RONS in the digestion of carnivorous plants, but generally indicated the involvement of RONS in proteolysis.
The techniques used by Chia et al. 2004 are very suitable for this topic (EPR spin trapping assay with DMPO, electrophoretic separations). Therefore from methodological point of view there is no basis for questioning them. The fact that no such research has been undertaken is also no the reason to criticize the work. In the work of Wal et al., the presence of ROS, specifically free radicals, was confirmed using a different technique (adrenochrome method).
The connection between ROS/RNS and carnivorous plants (food digestion) is a very rare topic. But seems to be interesting. Many studies have been published a long time ago, and not all of them confirm the results from a given publication. We confirmed Chia's results (Wal et al., 2022).
Reviewer 3 Report
Reviewer comments:
The manuscript presents a comprehensive analysis of the digestive processes in Nepenthes pitcher plants, drawing parallels with the human gastrointestinal (GI) system. The study focuses on the role of reactive oxygen and nitrogen species (ROS and RNS) in these digestive processes. The manuscript is generally well-written and provides valuable insights into the topic. However, there are some areas that require clarification and further elaboration.
1. The manuscript contains some long and complex sentences, which can make it challenging to follow the flow of the text. Simplifying these sentences or breaking them into shorter ones would enhance readability.
2. The title "Do RONS Modulate the Digestion in Pitcher Plants?" is relevant to the content of the manuscript. However, it's important to note that the title primarily suggests a focus on RONS (Reactive Oxygen and Nitrogen Species) and their role in pitcher plant digestion. Since the manuscript comprises discussions on various aspects of pitcher plant digestion, including the comparison with the human gastrointestinal system and the role of ROS and RNS, the title could be refined to reflect the broader scope of the content. A revised title might encompass both the role of ROS and RNS and the comparison with the human digestive system. This would give a more comprehensive overview of the manuscript's content and highlight the key elements of the review. For example: "Comparative Analysis of Digestive Processes in Nepenthes Pitcher Plants: Roles of ROS, RNS, and Parallels with the Human GI System."
3. The abstract could benefit from providing a concise statement of the review's objectives or specific research questions to give readers a clear idea of what to expect in the review article.
4. The first section on Carnivorous plants needs to be drafted with a proper flow following the title theme. For example: The origin and evolution of carnivorous plants, their diversity, then about their feeding systems and organ architecture, and lastly a glimpse of the various systems yet understood and yet unknown. Then set the stage on RONs, briefing why a review on RONs demands further studies to establish their role as digestive systems predominant in Pitchers.
5. Please revise the attached corrections (in MSWORD track changes), mostly suggested sentence rephrasing for clarity and other minor suggestions as well.
6. In the concluding section, consider summarizing the primary implications of your review for both the field and its potential applications. This will help readers understand the broader significance of your work.
7. If feasible, please supplement a section within human gut parallelism of pitcher traps with microflora features as well. Use these reads:
https://doi.org/10.1016/j.syapm.2015.05.006
8. Answer this: Most of the parallelism features have already been discussed an year before in a review you have cited in the paper: https://doi.org/10.1093/plphys/kiac232 . Then what new are you concluding beyond the role of RONs, which you only mentioned is yet under full establishment in the Pitchers?
Regards,
The other reviewer.

Minor English language editing (mostly sentence framing) is suggested. I believe the authors can manage it on their own.
Author Response
Thank you very much for your valuable comments, the answers are attached in the file.

Round 2
Reviewer 2 Report
Obviously, the authors and I have different ideas about how a review should be. Despite the authors' arguments, I do not see any significant new aspects in the manuscript that go beyond what is already known. For example, the point that the Freund et al (2022) review looked at more than one carnivorous species is actually a positive, not a negative, point.
I am not questioning the data of Chia et al (2004) or Wal et al (2022) that showed the existence of ROS/RNS in pitcher juice, but the mere proof of their existence is not sufficient to attribute a functional role to them.
I see this manuscript as a hypothesis paper at best, and the textbook portion should be significantly reduced.
I would like to positively emphasize the inclusion of the microflora and the change of title.
I leave it to the handling editor to follow my arguments or those of the authors.
Author Response
Thank you for the reviews.